# Transcriptional Profile Corroborates that *bml* Mutant Plays likely Role in Premature Leaf Senescence of Rice (*Oryza sativa* L.)

**DOI:** 10.3390/ijms20071708

**Published:** 2019-04-05

**Authors:** Delara Akhter, Ran Qin, Ujjal Kumar Nath, Jamal Eshag, Xiaoli Jin, Chunhai Shi

**Affiliations:** 1Department of Agronomy, Zhejiang University, Hangzhou 310027, China; 11516090@zju.edu.cn (D.A.); ranqin89@zju.edu.cn (R.Q.); jamaladam@zju.edu.cn (J.E.); jinxl@zju.edu.cn (X.J.); 2Department of Genetics and Plant Breeding, Sylhet Agricultural University, Sylhet 3100, Bangladesh; 3Department of Genetics and Plant Breeding, Bangladesh Agricultural University, Mymensingh 2202, Bangladesh; ujjalnath@gmail.com

**Keywords:** rice, *bml* (*brown midrib leaf*), transcriptional analysis, DEGs, GO, KEGG, KOG, cluster analysis

## Abstract

Leaf senescence is the last period of leaf growth and a dynamic procedure associated with its death. The adaptability of the plants to changing environments occurs thanks to leaf senescence. Hence, transcriptional profiling is important to figure out the exact mechanisms of inducing leaf senescence in a particular crop, such as rice. From this perspective, leaf samples of two different rice genotypes, the *brown midrib leaf* (*bml*) mutant and its wild type (WT) were sampled for transcriptional profiling to identify differentially-expressed genes (DEGs). We identified 2670 DEGs, among which 1657 genes were up- and 1013 genes were down-regulated. These DEGs were enriched in binding and catalytic activity, followed by the single organism process and metabolic process through gene ontology (GO), while the Kyoto Encyclopedia of Genes and Genomes (KEGG) pathway analysis showed that the DEGs were related to the plant hormone signal transduction and photosynthetic pathway enrichment. The expression pattern and the clustering of DEGs revealed that the *WRKY* and *NAC* family, as well as zinc finger transcription factors, had greater effects on early-senescence of leaf compared to other transcription factors. These findings will help to elucidate the precise functional role of *bml* rice mutant in the early-leaf senescence.

## 1. Introduction

Leaf senescence is the ending period of leaf growth in the vegetative cycle, and acts as an indicator of physiological maturity of the harvest product. However, premature leaf senescence may mislead the growers to harvest too early. Many reports have pointed out that it plays adaptive roles in plants in changing environments by fine tuning the regulatory genes [1,2]. The majority of these genes are involved in (1) recycling and re-using the nutrients from senescent leaves for newly developed organs, and (2) enhancing tolerance against biotic or abiotic stresses [3,4,5].

Leaf senescence is a vital agronomic attribute for crop yield and quality [6,7,8]. A wide range of physiological processes occur in the leaves during the process of senescence. The rate of assimilation reduces more rapidly with the increase of catabolism, such as degradation of the chloroplast, reducing photosynthesis, and the degradation of macromolecules [9]. The senescence process might also be affected by internal as well as the external factors. The latter case is environmental conditions, e.g. soil moisture, light, temperature, nutrient content and pathogens [10,11]. Therefore, it is important to understand the internal factors that are responsible for leaf senescence, as this knowledge could help to elucidate the basic molecular mechanism. Exploration of that mechanism will provide a means for controlling leaf senescence in crop plants [2].

Many studies have been carried out on senescence-associated genes (SAGs) in different plant species [9,12,13,14], and currently, the database of leaf senescence comprises some 130 SAGs [15]. However, the findings have not provided a complete explanation of the molecular mechanisms of leaf senescence in monocots like, rice, wheat, maize, sorghum and barley. It has been established that the chlorophyll (Chl) metabolism and degradation pathways are involved in several SAGs of leaf senescence [16]. During chlorophyll degradation, Chlb is changed to Chla in the first leaf, wherein *NYC1* [17] and *NOL* [18] are involved in chlorophyll-b degradation. In *A. thaliana*, the gene *PAO* (*Pheophorbide A Oxygenase*) takes part in the degradation of chlorophyll, leading to leaf senescence [19]. *PPR* (pentatricopeptide repeat) genes control chloroplast gene expression, and eventually, senescence of leaves in rice [20]. Plant hormones like MeJA (methyljasmonate) and JA (jasmonate) have been recognized as leaf senescence enhancing substances for rice [21]. The gene *OsCOL1b* (*CORONATINE INSENSITVE 1b*) encodes the JA receptor of *Arabidopsis*, and the mutant *Oscoi1b* delays leaf senescence and shows insensitivity to MeJA [22]. Two important genes, *OsPME* and *OsTSD2* [23], encode pectin-esterase and methyltransferase respectively were identified through mGWAS (metabolite-based genome-wide association study) [23]. Besides JA, IAA also plays a complex role in leaf senescence regulation. Leaf senescence can be delayed by exogenous IAA application [24]. Leaf senescence could be positively regulated by auxin-responsive genes such as SAUR (small auxin-up RNA) [24,25,26,27]. The gene *OsPLS1* encodes a vacuolar H^+^-ATPase subunit A1, and negatively regulates leaf senescence at the beginning [28].

Leaf senescence in rice is regulated by several transcription factors (TFs); many of them belong to WRKY transcription factor [29,30]. In Arabidopsis, the *AtWRKY70*, *AtWRKY53*, *AtWRKY54* genes are engaged in the signaling pathway of SA [31] which induces cell death and leaf senescence [30,31,32]. The over-expression of *OsWRKY42* exhibits ROS accumulation and accelerates leaf senescence through repressing *OsMT1d* in rice [33]. The chlorophyll metabolic pathway involved several GATA-zinc finger TFs, like *GNC* (*GATA nirate-inducible carbon-metabolism*) [34], degrades the chlorophyll in *A. thaliana* [34,35]. The over-expression of *OsGATA12* reduces degradation of chlorophyll and delays leaf senescence, reducing the number of leaves and tillers, as well as enhancing yields in rice [36]. In addition, the *GPC-B1* (*grain protein content-B1*) and *OsNAC106* (*N-acetylcysteine106*) genes are senescence-linked transcription factors [37]. Among them, *GPC-B1* accelerates leaf senescence [37], while *ONAC106* regulates leaf senescence inversely [38]. The transporter family genes *OsSWEET5* (*sugar transporter family5*) [39] and *Yr18/Lr34/Pm38* from the ABC-transporter family also induce leaf tip senescence [40] in rice.

Transcriptomic analysis will help to identify the genes associated with leaf senescence, because the transcriptional profiling of the leaf senescence genes in rice is still under exploration. Therefore, the identification of these genes through RNA-Seq (RNA sequencing) is crucial to understanding the molecular mechanisms of leaf senescence and to developing stress-adaptive rice cultivars in order to achieve higher yields. RNA-Seq is a quantification technology of transcripts which provides precise levels of transcripts measurements; therefore, it has been successfully and widely used for profiling transcripts with annotations, and in the identification of genes in different plant species [41,42,43,44,45,46].

In the present study, RNA-sequencing analysis was accomplished with a mutant of early-leaf senescence, *bml* mutant, and its WT variety, Zhenong 34, at an early-tillering stage. In our previous study, this *bml* mutant was identified and reported in the mutational rice population developed by ethyl methane sulfonate (EMS) mutagenesis. Such a mutant was differentiated by the phenotypic character of the early-senescence in leaf from the M_2_ population [47]. Herein, we performed transcriptomic profiling of that *bml* mutant Vs WT to identify the differentially-expressed genes and post-translation modifications in order to elucidate the mechanisms involved in leaf senescence. GO enrichment, Kyoto Encyclopedia of Genes and Genomes (KEGG) pathway and KOG/COG (Clusters of Orthologous Groups) analyses were also performed at an early-tillering stage. This transcriptome data will provide information necessary to understand leaf senescence in rice.

## 2. Results

### 2.1. Phenotypic Characterization 

No observable differences were found between the *bml* mutant and WT in the seedling to tillering stage [47]. Senescence was found to start in the bottom leaves at the early-heading stage in the *bml* mutant, wherein the other leaves then became yellow with progressive senescence (Figure 1A). The midrib of the leaves in the *bml* mutant started to turn brown, and lesion phenotypes appeared on the leaf blade, gradually covering the whole leaf (Figure 1A,B). Leaf senescence was accelerated more in the *bml* mutant during the late-heading and grain filling stages, compared to WT. Our previous study showed that the *bml* plants demonstrated slower growth, with significantly reduced numbers of tillers, shorter panicle lengths, lower seed-settings, fewer grains in the panicles, lower 1000-grain weights, and smaller seed sizes (length and width) compared to WT [47]. 

Moreover, genetic analysis indicated that the early-senescence of leaf was controlled by a single recessive gene [47]. The mutants showed abnormal leaf cells with degraded chloroplasts and less chlorophyll, a reduced photosynthetic rate (P_n_), less stomatal conductance (Gs) and intercellular CO_2_ concentration (Ci), reduced transpiration rate (Tr) and Fv/Fm (maximal quantum yield of PSII) compared to WT at the heading stage [47]. 

### 2.2. RNA-Seq Analysis

RNA-seq was performed for the transcriptomic profiling of *bml* mutant using leaf samples collected from three replicates of *bml* mutant and WT plants. The cDNA of WT and *bml* leaf samples were sequenced using Illumina technology to determine the probable DEGs and the pathways involved in the process of early-leaf senescence in the *bml* mutant. On average, 60,545,108 bp and 62,982,574 bp raw reads were retrieved from the RNA-seq data for the *bml* mutant and WT, respectively (Table 1). After removing the adaptor, ambiguous nucleotides, and low-quality sequences, approximately 56,637,709 bp and 61,147,444 bp clean reads were obtained from the WT and *bml* mutant, respectively, which covered an average length of 147.1 to 147.5 bp. The GC content of the clean reads ranged from 54.71 to 55.60% in different libraries, and the Q30 percentage exceeded 94.9% (Table 1).

A total of 110,715,792 clean reads were generated by sequencing six cDNA libraries. All reads were classified into three categories: total mapped, multiple mapped and uniquely mapped. Of all the reads, 94.01–94.27% read were totally mapped, 91–93% were uniquely mapped, and 2–3% were mapped in the genome (Table 2). A summary of the sequencing data, alignment statistics of the reads to the reference gene, alignment statistics to reference genome and QC items for each sample are given in Table 1 and Table 2.

A correlation of a heat map was constructed using the correlation coefficients of replicated samples of the *bml* mutant and WT (Figure 2). The correlation coefficient among *bml* samples was 0.95 (bml1, bml2 and bml3), while that among the WT samples was 0.94 (WT1, WT2 and WT3), which had a tendency for unity. These results revealed that the *bml* group (*bml1*, *bml2* and *bml3*) differed significantly from the WT group (*WT1*, *WT2* and *WT3*).

The average numbers of SNPs were 96,368 and 96,228 in the *bml* mutant and WT respectively, whereas *INDEL*s were 10,861 and 11,020, respectively. SNP-affected regions were divided into two categories, i.e., genic (UTR and CDS) and inter-genic regions. A major part of the SNP-affected regions was covered by the genic region for both the *bml* mutant and WT (Appendix A). Two kinds of nucleotide substitutions were identified at homozygote polymorphic SNPs due to transversions (A/C, A/T, C/G, G/A and G/T) and transitions (A/G, G/A, T/C and C/T). The occurrence of SNPs in transitions was more frequent for both the *bml* mutant and the WT (Appendix A).

### 2.3. Identifications of DEGs

The DEGs between *bml* mutant and WT were figured out at the early-tillering stage to explore the genes involved in leaf senescence of *bml* mutant (Appendix A). A volcano plot of all genes was drawn to characterize the distribution of DEGs in the selection at threshold level (Figure 3A). A total number of 2670 DEGs was identified, among them 1657 genes were up-regulated and 1013 genes were down-regulated in *bml* Vs WT leaf. Moreover, we also detected the specific DEGs by the Venn diagrams and found 1858 genes as commonality in all the samples (Figure 3B).

### 2.4. GO Enrichment Analysis

Gene Ontology (GO) is an international standardized gene functional classification system that defines the genes according to three terms: molecular function, cellular component and biological process. The enriched GO terms of down- and up-regulated DEGs was analyzed and presented as bar-plot. A total 16,510 genes was assigned in GO and 1503 (9.10%) DEGs annotated significantly in the GO database, among them 537 and 966 genes were regulated down- and up-, respectively. These genes were assigned to 50 subcategories (Figure 4 and Appendix A). Among them, a majority of the genes were categorized as biological process (44.89%) followed by molecular (32.65%) and cellular functions (24.48%) (Figure 4 and Appendix A). A higher proportion of the genes were involved in biological function because the leaf senescence is a physiological process, which could bring the changes in many biological processes and regulated by the genes and perceived the signals in the early-tillering stage for enhancing the leaf senescence at early heading stage. In the category of biological process, the majority of the genes were assigned in the metabolic subcategory followed by single organism and cellular process. A significant proportion of the clusters were classified as binding, followed by catalytic activity with transcription factor activity and sequence specific DNA binding but no RNA binding in the molecular function category (Figure 4).

For cellular component subcategory, the largest portion of the genes was subcategorized as the involvement in membrane followed by cell with cell part, membrane part and organelle part. A few genes were assigned to the rhythmic process, immune system process, extracellular matrix and mettalochaperon activity categories. The scatter plot of the DEGs (up- and down-regulated) also showed that maximum number of DEGs involved in binding and catalytic activity followed by the single organism process and the single organism metabolic process (Appendix A).

### 2.5. KEGG and KOG Pathway Analysis of DEGs

KEGG analysis with DEGs was executed for further investigating the pathway involved in leaf senescence of the *bml* mutant at the early-tillering stage. The unigenes orientations of metabolic pathways were investigated through bar-plot analysis in the KEGG database. Among 5226 assigned genes, only 527 genes were found to be involved in five pathways, such as the cellular, environmental and genetic information processing as well as in the metabolism and organismal systems (Figure 5A and Appendix A).

The scatter plot analyses of the significant KEGG enrichment showed the function and highly differentially expressed genes are represented by the size of the dots upon each function (Figure 5B and Appendix A). Here, we selected only top 30 KEGGs based on the highest level of KEGG enrichment. Among them, ko04075 (plant hormone signal transduction), ko01230 (biosynthesis of amino acid), ko00940 (phenylpropanoid biosynthesis), ko00710 (carbon fixation in photosynthetic organisms), ko00195 (photosynthesis) and ko00480 (glutathione metabolism) DEGs had significant enrichment in KEGG pathways related to the leaf senescence. It also indicated that the genetic signal of early leaf senescence might be perceived in the earlytillering stage of *bml* mutant plants through alteration of hormonal signal transduction and photosynthesis. 

The most relevant biological pathways for early-leaf senescence of the *bml* mutant could be identified by enrichment analysis. Functions of uniqueness were classified according to COG/KOG databases using orthologous gene products. A total of 10106 non-redundant uni-genes was found in KOG analysis, among which 929 genes showed differential expression. Unique DEGs were divided into 26 KOG sub-functional groups (Figure 6); it was determined that the maximum of them was involved in the signal transduction mechanism followed by general function prediction, post-translational modification, protein turnover and chaperone with secondary metabolite biosynthesis, transport, as well as catabolism. Few unique DEGs were found to involve in cell motility.

### 2.6. Confirmation of Gene Expression Profile through qRT-PCR

The qRT-PCR was executed to validate the RAN-seq data using 25 randomly selected genes relevant to leaf senescence (Table 3). The gene expression pattern was similar to the transcriptome data of RNA-seq (Table 3), indicating the consistency and reliability of the transcriptomes.

*bml* showed the senescence and lesion mimic phenotype and functions related to hormone signaling and photosynthesis pathways and transcription factor genes. In this regard, we prioritized the identification of the differentially-expressed genes involved in early-leaf senescence (Table 3). In the early-leaf senescence of rice mutant *bml*, DEGs were enriched in plant hormone signal transduction, photosynthesis, and photosynthesis with antenna proteins pathways including *WRKY*, *NAC*, *GATA* transcription factors, chlorophyll metabolism as well as different biological processes. We also identified 22 DEGs orthologs of the rice genes related to abscicic acid (ABA; 6 genes), indole acetic acid (IAA; 6 genes), jasmonic acid (JA; 2 genes), and salicylic acid (SA; 8 genes) responses (Appendix A). Among them, the orthologs of *IAA26* (*LOC_Os02g57250*) and *IAA13* (*LOC_Os01g09450*) were down-regulated in the *bml* mutant. The relative expression of the genes of ABA signal transduction was increased in *bml* compared to WT. *PR1* family-related genes are involved in the SA signaling pathway; therefore, we found that SA-related *LOC_Os01g28450*, *LOC_Os01g09800* and *LOC_Os07g03730* genes were significantly up-regulated in the *bml* mutant compared to WT. JA signaling genes, e. g. *JAZ10* (*LOC_Os03g08330*) and *JAZ5* (*LOC_Os04g32480*) were also up-regulated in the *bml* mutant (Table 3). The results confirmed that the gene expression was matched and steady with the transcript analysis accomplished through RNA-seq. 

### 2.7. Cluster Analysis of DEGs

We performed a cluster analysis using the expression values of the DEGs to identify the genes directly connected to leaf senescence. Hierarchical clustering was performed considering 10% cut-off of the tree constructed from DEGs by the R package (Figure 7).

In total, 4101 genes which exhibited diverse expression patterns between the *bml* mutant and WT were classified into 10 sub-clusters, (Figure 8 and Appendix A). Six clusters showed the changes of gene expression, while 4 exhibited genes with up-regulation, and 2 contained genes with down-regulation in *bml*. We found that the genes presented in the clusters were associated with senescence, and most genes showed up-regulation in the *bml* mutant. Among the up-regulated genes, 139 encoded kinase family proteins, and 19 belonged to receptor-like genes. Additionally, 55 genes were annotated as transcription factors or DNA-binding factors, 17 with *WRKY* transcription factors, while the rest showed orthology to *bZIP*, *NAC* and zinc finger transcription factors (Appendix A).

## 3. Discussion

A rice mutant, *bml,* showed the effect on plant growth and development. By genetic analysis, it was concluded that early-leaf senescence was governed by the recessive *bml* gene [47]. RNA-Seq analyses were performed to determine whether the changes of the transcriptome of senescence-associated genes, rather than solely the *bml* recessive gene, started at the beginning of tillering, by comparing the sequence data of *bml* and WT of rice at early-tillering stage. It was kept in mind that RNA-Seq analysis helps with precise the transcript profiling, annotation and identification of genes in plants [41,43,46].

Most of the correlation coefficients of our *bml* samples were 0.95, while that of the WT samples was 0.94, which were approaching unity, indicating a higher level of similarity in the expression patterns among the samples [45]. These results revealed high sample repeatability, and reflected significant differences between the *bml* and WT groups.

A total of 25 genes were randomly selected from the identified DEGs for testing of their expression at the early-tillering stage by qRT-PCR. The qRT-PCR results showed that the genes were expressed differentially, and they confirmed the trustworthiness of our transcriptome data. The identification of DEGs in diverse transcripts was very important for finding the differentially-expressed genes between the samples, and for carrying out an additional functional analysis of the outcomes [45,48]. Among the 2670 DEGs, 1657 and 1013 genes showed up- and down-regulation respectively. About 50 enriched GO terms were detected by the GO enrichment analysis of the DEGs at the early-tillering stage. Moreover, KOG analysis of DEGs showed the involvement of uni-genes in the signal transduction mechanism, which is relevant to early-leaf senescence, and which was recognized at the early-heading stage; these results were confirmed by Zhang et al. [32]. Mainly, the WRKY transcription factor genes were significantly up-regulated in the *bml* mutant, indicating their direct involvement in early-leaf senescence. Similarly, the orthologous genes of *AtWRKY40*, *AtWRKY53* and *AtWRKY70* transcription factors were engaged in SA signaling transduction pathways in leaf senescence [31,32,49]. 

The functional activity of WRKY transcription factors occurs in the upstream of the regulatory network of leaf senescence. By comparing the expression profiles of the differential genes of *bml* mutant and WT, we deduced that the gene encoded protein kinase *WRKY* transcription factor, *GATA* transcription factors, *bZIP*, *NAC* and zinc finger transcription factors play potential roles in the leaf senescence process, which is in agreement with a previous study by Leng et al. [5]. We also found that down-regulated DEGs were enriched in GO terms such as photosystem I, photosystem II and chlorophyll metabolism at the early-tillering stage (Appendix A); those were reported as the factor of degenerative phase in earlier studies [50]. On the process of senescence, the leaf cells pass through a remarkable transition and systematic deterioration of different cellular structures [29]. We found several differentially-expressed genes in the *bml* mutant related to chlorophyll metabolism and which contribute to leaf senescence. 

The expression of orthologs of chloroplast precursor-relative and TPR (tetratricopeptide repeat) like super family genes *LOC_Os08g09270*, *LOC_Os06g02120* and *LOC-Os01g25600* were down-regulated in the *bml* mutant (Appendix A). In the *bml* mutant, the genes of grana degradation, photosynthesis pathway, photosystem I and photosystem II (Photosynthesis - antenna proteins pathway) were shown to be down-regulated, which supports our results by the down-regulation of *NYC1*, *ATFD3*, *ATLFNR2*, *LHCA5* and *LHCA2* genes [18,51,52] which causes early-leaf senescence. The orthologs of *ATCOL2* (a *GATA* transcription factor) were down-regulated, which can influence leaf senescence by regulating chlorophyll accumulation [35,36]. Pentatricopeptide repeat (PPR) super family proteins can affect the content of chloroplast through regulating the gene expression of the chloroplast, e.g., *OspTAC2* gene in rice [20] and other PPR, *PPR53* genes might also change the gene expression of chloroplast and influence the development of leaf in maize [53]. We found several PPR-like genes in our transcriptome profiles, such as *LOC_Os01g48380*, *LOC_Os06g07550*, *LOC_Os10g10170* and *LOC_Os06g09880*, which were down-regulated in the *bml* mutant and might play a role in the process of early leaf senescence in rice.

KEGG pathway analysis showed that leaf senescence is affected by the plant hormone signaling pathway. A prior study reported that SA, JA, ABA and ET accelerate leaf senescence, while IAA, GA and CK were found to delay its development [54]. In the present study, we found IAA related genes, such as the *LOC_Os02g57250* and *LOC_Os01g09450* as orthologs of *IAA26* and *IAA13*, respectively, were down-regulated in the *bml* mutant. The expression of ABA hormone signal transduction pathway genes increased in the *bml* mutant. The *PR1* family-related genes, such as *LOC_Os01g28450*, *LOC_Os01g09800*, and *LOC_Os07g03730* involved in SA signaling pathway showed significant up-regulation in the *bml* mutant. JA signaling genes, such as *JAZ10* (*LOC_Os03g08330*) and *JAZ5* (*LOC_Os04g32480*), also showed up-regulation in the *bml* mutant (Table 3), which should enhance leaf senescence. These results revealed that leaf senescence in the *bml* mutant might be controlled by different hormones, as stated in a previous study of wheat [32].

The cluster analysis of DEG expression patterns gave us an idea and the opportunity to classify and identify senesce-associated genes (SAGs) in rice. We gave more emphasis to clusters in which gene expression patterns showed obvious up- and/or down regulation in the *bml* mutant compared to WT (Figure 8 and Appendix A). Many types of protein kinases genes, like receptor-like protein kinase, transcription factors and transporter proteins, were significantly enriched. The number of *WRKY* family, *NAC* and zinc finger transcription factor genes was higher than that of other transcription factor genes. A previous report revealed that many transcription factor genes were connected to leaf senescence in *Arabidopsis* [30], and might play significant roles in the early-senescence of rice. So, we concluded that signal transduction and substance transport were associated, and that the perceived the genetical signals at the early-tillering stage of *bml* rice plants contribute to early leaf senescence. 

We explored and profiled the transcriptomes of *bml* mutant and WT plants at the early-tillering stage to determine and annotate the transcripts linked to leaf senescence. The GO enrichment, the KEGG pathway of DEGs and the KOG/COG were analyzed at the early-tillering stage. Multiple pathways were found to be involved in the leaf senescence process in rice. Among them, plant hormone signal transduction, photosynthesis, photosynthesis antenna protein pathways played predominant roles in the early leaf senescence process. These pathways’ genes might be evacuated in chlorophyll break down or degradation; the accumulation of antioxidant activity and ROS molecules play a direct role in leaf senescence. Our results confirmed the transcriptional factor and signal transduction genes which bring about changes in the regulation of different hormones which accelerate the early senescence process of leaf in rice. Therefore, in our future breeding programs, we could emphasize those genes to ensure delayed senescence by increasing the duration of photosynthesis and higher yields in rice.

## 4. Materials and Methods

### 4.1. Plant Materials

Two rice genotypes, i.e., the wild form of Zhenong 34 and its mutant of *brown midrib leaf* (*bml*) from *Oryza sativa* L. ssp. *Indica*, were used for RNA-sequencing. The *bml* is the mutant of Zhenong 34 produced by ethyl methane sulfonate (EMS) mutagenesis. The mutant plants were selected by gene mapping according to a bulk segregant analysis (BSA) method, as well as being differentiated phenotypically and recognized as early-senescence of leaf compared to WT in the M_2_ population [47]. Plants were grown in rice fields at the Zhejiang University in Hangzhou, China (30°15′49″ N, 120°7′15″ E), during March–July 2017; the average temperature was 14 °C to 33 °C, with 60 to 70% R.H. The leaf tissues were collected from the plant seedlings (25 days after sowing and just before transplanting), early tillering (15 days after transplanting) and early booting (45 days after sowing) stages, 10 leaves from 10 plants at same growing stage of the WT and *bml* plants were collected as biological replicates and pooled together before being immersed in liquid nitrogen. All samples were stored in a refrigerator at −80 °C until RNA extraction. 

### 4.2. RNA Isolation, Library Preparation and Sequencing

Total RNA was extracted using the total RNA Extractor (Trizol) kit (B511311, Sangon, China) according to the manufacturer’s protocol. The quality and quantity of the extracted RNA were assessed using Agilent 2100 Bioanalyzer (Agilent Technologies, CA, USA) and NanoPhotometer^®^ spectrophotometer (IMPLEN, CA, USA). High-quality RNA samples were subsequently submitted to the Sangon Biotech Co. Ltd., Shanghai, China for sequencing and library preparation. 

The mRNA was purified from the total RNA using poly-T oligo-attached magnetic beads. Fragmentation was carried out using the divalent cations under an elevated temperature in a VAHTSTM First-Strand Synthesis Reaction Buffer (5×). First strand cDNA was synthesized using a random hexamer primer and M-MuLV Reverse Transcriptase (RNase H-). Second strand cDNA synthesis was subsequently performed using DNA polymerase I and RNase H. The remaining overhangs were converted into blunt ends via exonuclease/polymerase activities. After adenylation of the 3´ ends of the DNA fragments, the adaptor was ligated in order to prepare the libraries. The adapter primer sequences were used as the forward primer: 5’AGATCGGAAGAGCACACGTCTGAAC3´, and the reverse primer: 5´AGATCGGAAGAGCGTCGTGTAGGGA3´. The library quality was assessed on the Agilent Bioanalyzer 2100 system (Agilent Technologies). A total of six libraries, including three from the *bml* mutant and three from WT, were generated. Paired-end sequencing of the libraries was performed using an Illumina HiSeqTM2000 (Illumina, San Diego, CA, USA). 

### 4.3. Data Filtering, Reads Mapping and RNA-Seq Data Analysis

Raw reads of primary sequencing were collected from the Illumina HiSeqTM 2000 and filtered (before downstream analysis for reducing data noise) to retrieve the clear reads. During the filtration of the data, adapter, poly-N, as well as low-quality reads were removed from the raw data to ensure the reliability of the analysis. The clean reads were stored in the FASTQ format [55], and Bowtie 2 [56] was used to map clean reads to the reference gene. HISAT2 (version 2.1.0) [57] was used with the parameters: hisat2-mm-dta-p 10-x genome.fa-known-splicesite-infile genes.ss-1 R1.fastq.gz-2 R2.fastq.gz to align the clean reads with the reference genome of the Nipponbare subspecies of rice (Oryza_sativa.v7.0; http://rice.plantbiology.msu.edu/) with an ideal match or one mismatch. RSEM, a precise and accessible software tool, was used to determine the transcript isoforms of the same gene [58]. 

### 4.4. Transcriptomic Analysis 

DESeq2 (version 1.12.4) was used to determine the DEGs between two samples. Genes were considered as differentially- and significantly-expressed if *q*-value *<* 0.001 and |Fold Change| *>* 2. Expression values of zero for the samples were treated as 0.01 during normalization, because zero could not be plotted onto the log plot. 

Next, DEGs were mapped to the GO terms (biological functions) in the database, the number of genes was calculated in every term, and a hypergeometric test was performed to identify any significantly enriched GO terms for the genes list in the background of the reference list. KEGG pathway analysis identified significantly enriched metabolic pathways and signal transduction pathways in DEGs compared to a reference genome. 

### 4.5. Quantitative Real-Time PCR

For RNA-seq data validation, 25 genes were randomly selected from DEGs and tested for their expression levels at the early-tillering stage using quantitative real time PCR (qRT-PCR). Gene-specific primers were designed using the Primer 5 software (Appendix A), with the parameters of 100–500 bp PCR product size, 20 ± 2 bp primer length, melting temperature 50–62 °C, GC content 40–60% and run in automatic search mode. The qRT-PCR was performed in the Roche light cycler 480 real-time system (Roche, Germany) using Ex TaqII (Takara, Tokyo, Japan) according to the manufacturer’s instructions, with five technical replicates of 10 μL reaction mixture. Then, 10 μL reaction mixture was mixed with 5µl 2× quanti speed SYBR mix with 1 µL (10 pmol) for each of the forward (F) and reverse (R) gene specific primers, 1 µL template cDNA (50ng) and 2 µL distilled-deionized water (ddH_2_O). qRT-PCR was performed following denaturation at 95°C for 30s, 40 cycles of denaturation at 95 °C for 5 s, annealing at 55 °C for 20 s, and extension at 72 °C for 10 s. To calculate the expression, the *OsActin* reference gene was used as an internal control [59]. The relative expression was calculated by the 2^-ΔΔCT^ method [60] using the cq values normalized by the cq of the rice *OsActin* gene.

## Figures and Tables

**Figure 1 ijms-20-01708-f001:**
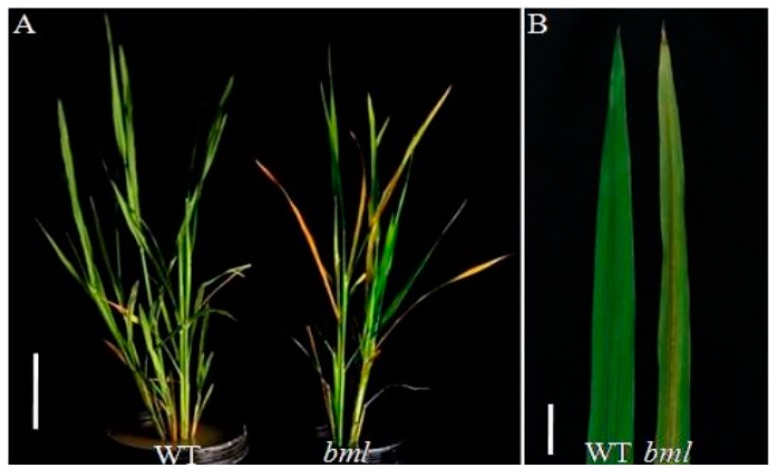
Phenotypes of bml mutant and WT rice plants. (**A**). Phenotype observed at the heading stage in bml mutant compared to WT. Bar = 10 cm. (**B**). Magnified view of the part of flag leaf from the top of the plant displayed in A. Bar = 5 cm.

**Figure 2 ijms-20-01708-f002:**
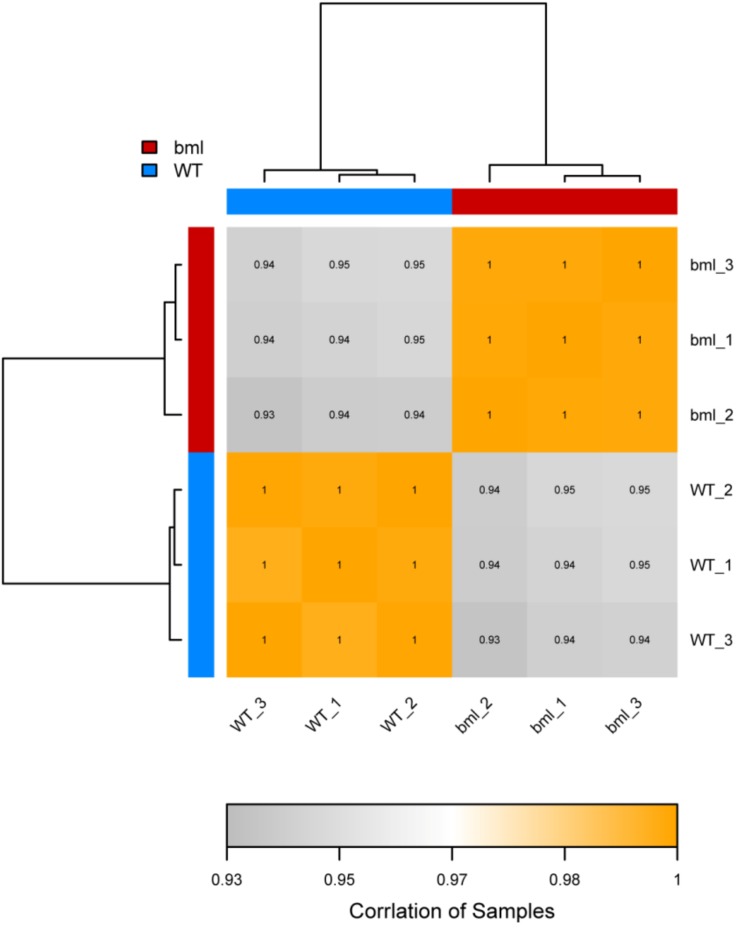
Heat-map diagram of inter-sample correlation analysis. The color block represents the correlation index value. The grayer the color, the lower the correlation index between samples and higher intensity of the color represents the higher correlation coefficient.

**Figure 3 ijms-20-01708-f003:**
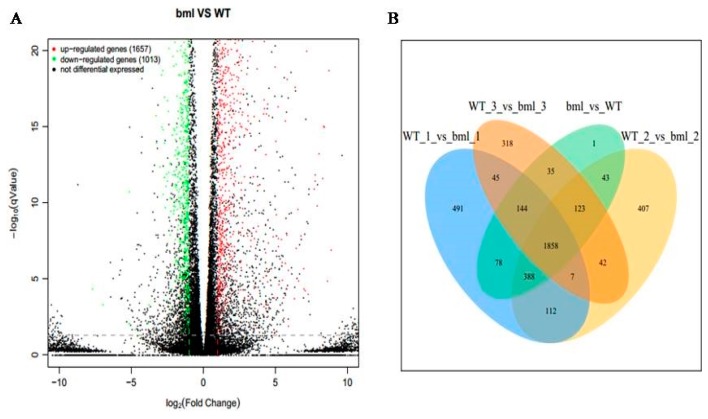
Volcano map and Venn diagram of DEGs. (**A**)**.** DEGs are shown by volcano map between *bml* and WT at early-tillering stage. The horizontal axis is the fold-change (log_2_(B/A)) value of the differential expression of the gene in different groups of samples, and the vertical axis is the statistical significance level with p value representing the changes in gene expression. Each point in the figure represents a gene, in which red indicates up-regulated genes, green indicates down-regulated genes, and black indicates non-differentiated genes. (**B**)**.** DEGs between *bml* mutant and WT at early-tillering stage are plotted by Venn diagram. The comparison between different groups is represented by different colors, and the numbers in the figure represent the number of specific or shared DEGs. The overlap region indicates the number of differentially expressed genes shared by different comparison groups, and the non-overlapping region indicates the number of DEGs unique between the different comparison groups.

**Figure 4 ijms-20-01708-f004:**
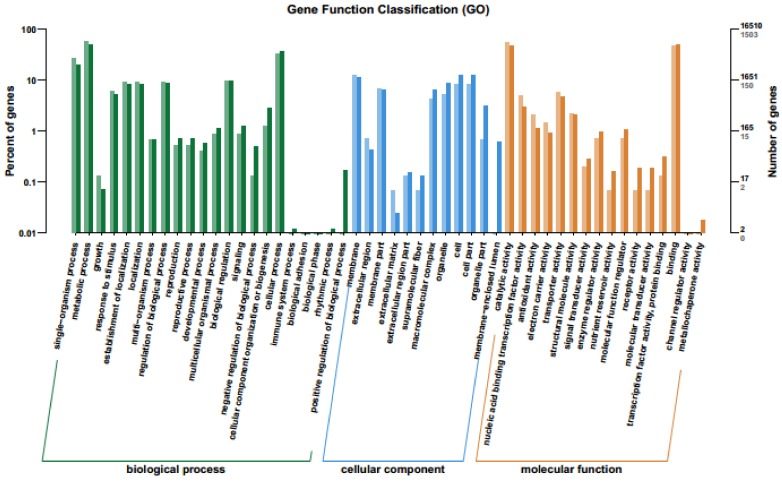
Differential gene ontology (GO) analysis of functional enrichment with up- and down-regulated DEGs between *bml* mutant and WT at early-tillering stage. The horizontal axis is the functional classification and the vertical axis is the number of genes in the classification (**right**) and the percentage of the total number of genes on the annotation (**left**). Different colors represent different classifications. Light colors on the histogram and axis represent differential genes, and dark colors represent all genes.

**Figure 5 ijms-20-01708-f005:**
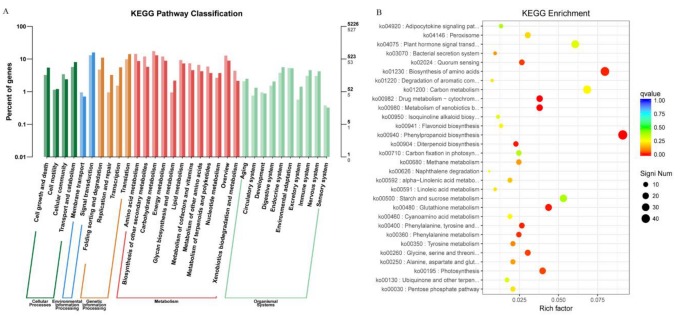
KEGG (Kyoto encyclopedia of genes and genomes) analyses of DEGs between *bml* and WT at early-tillering stage by barplot and scatter diagram. (**A**). KEGG analysis of DEGs through barplot. The horizontal axis is the functional classification, and the vertical axis is the number of genes in the classification (right) and the percentage of the total number of genes on the annotation (left). Different colors represent different classifications: light colors on the histogram and axis represent differential genes, and dark colors represent all genes. (**B**). Scatter plot of significant enrichment function. The vertical axis represents functional annotation information, and the horizontal axis represents the function of the Rich Factor (the number of differential genes annotated to the function divided by the number of genes annotated to the function). The size of the Q value is represented by the color of the dot; the smaller the Q value, the closer the color is to red, and the more the differential genes contained under each function are represented by the size of the dots. Only the top 30 GOs with the highest levels of enrichment were selected.

**Figure 6 ijms-20-01708-f006:**
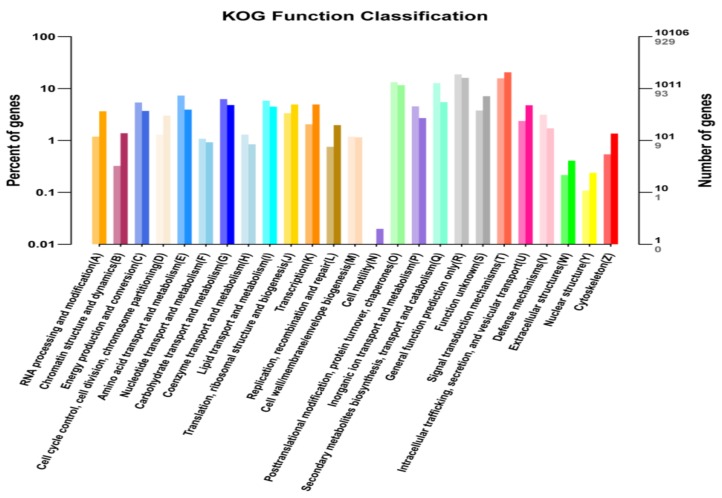
KOG/COG (Clusters of Orthologous Groups) analyses of DEGs between *bml* mutant and WT at early-tillering stage. The horizontal axis is the functional classification, and the vertical axis is the number of genes in the classification (**right**) and the percentage of the total number of genes on the annotation (**left**). Different colors represent different classifications. Light colors on the histogram and axis represent differential genes, and dark colors represents all genes.

**Figure 7 ijms-20-01708-f007:**
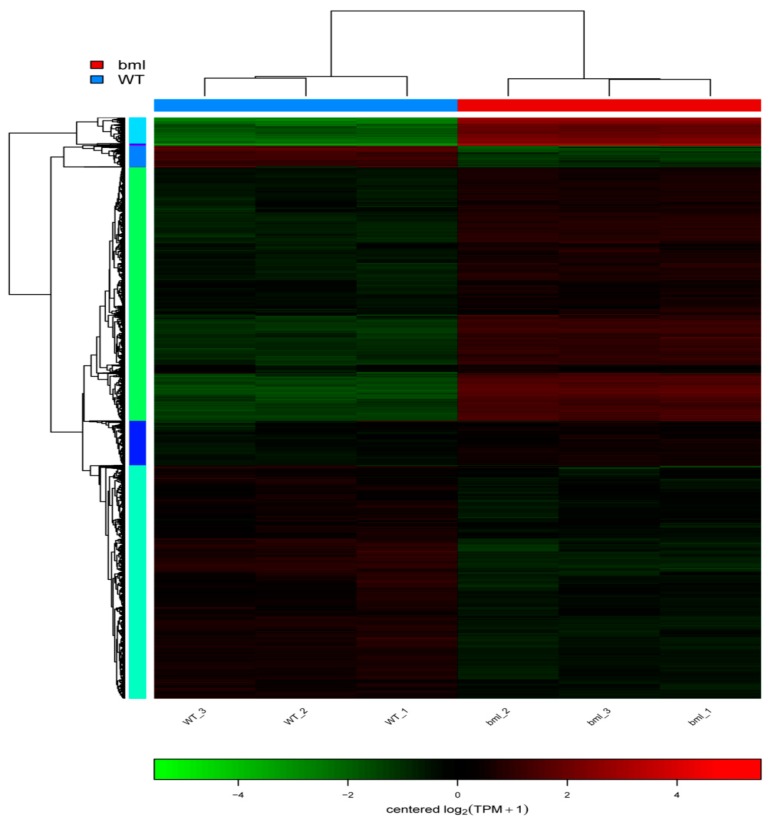
Clustering heat map of differential gene expression. Each row in the Figure represents a gene, each column represents a sample, and the color indicates the amount of expression of the gene in the sample. Red indicates that the gene is expressed in the sample, and green indicates that the expression level is low. The left side is a tree diagram of gene clustering. The closer the two gene branches are, the closer they are. The upper part is the tree diagram of the sample cluster, and the lower part is the name of the sample; the two samples are separated. The closer it is, the closer the expression patterns of all the genes in these two samples are, i.e., the closer the trend of gene expression changes.

**Figure 8 ijms-20-01708-f008:**
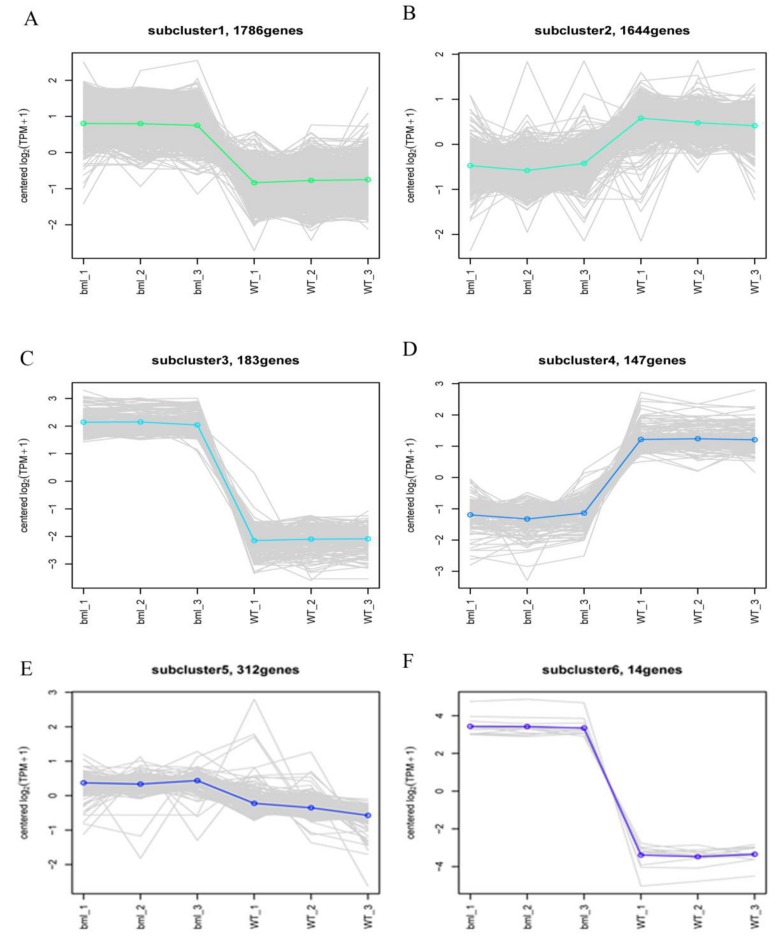
Differential gene module expression trends by the line chart. The expression trend line graph of each sub-module, the horizontal axis is the sample, and the vertical axis is the expression level of the gene in the sample. Each line in the figure represents a gene, and the colored lines represent the mean of a set of genes. Each graph shows one type of expression pattern, a trend that reflects changes in the expression levels of this group of genes; (**A,C,F**) up-regulated pattern of gene in the *bml* mutants but down-regulated in WT, (**B,D**) up-regulated pattern of gene in the WT but down-regulated in *bml* mutants and (**E**) genes had no response due to mutation.

**Table 1 ijms-20-01708-t001:** Reads summary of samples by Illumina deep RNA sequencing.

Sample Name	Raw Reads	Average Raw Read Length (bp)	Clean Reads	Average Clean Read Length (bp)	Clean Base (bp)	Error (%)	Q20 (%)	Q30 (%)	GC Content (%)
WT	60,545,108	150	56,637,709	147.10	30,965	0.00	98.44	94.64	54.71
*Bml*	62,982,574	150	61, 147, 444	147.50	43,180	0.00	98.59	94.99	55.60

**Table 2 ijms-20-01708-t002:** Alignment statistics summary of RNA-seq and mapping results.

Sample Name	Total Reads	Total Mapped (%)	Multiple Mapped (%)	Uniquely Mapped (%)	Non-Splice Reads (%)	Splice Reads (%)	Reads Mapped in Proper Pairs (%)
WT	53579398	94.01	3.00	91.00	61.46	29.55	87.31
*bml*	57136394	94.27	2.56	93.45	63.36	28.33	88.06

**Table 3 ijms-20-01708-t003:** Relative expression of 25 randomly-selected genes for comparison between the *bml* mutant and WT group with respect to RNA- seq and qRT PCR.

Gene ID	Gene Name	*bml* vs. WT		Gene Description	Pathway/Function Involved
		RNA-Seq log_2_ Fold Change	qRT PCR Fold Change		
*LOC_Os01g12710*	*NYC1*	−1.509	−0.880 ± 0.02	NAD(P)-binding Rossmann-fold superfamily protein	Porphyrin and chlorophyll metabolism
*LOC_Os02g52650*	*LHCA5*	−1.266	−569 ± 0.207	Photosystem I light harvesting complex gene 5	Photosynthesis-antenna proteins
*LOC_Os09g26810*	*LHCA2.1*	−1.0265	−0.879 ± 0.0890	Photosystem I light harvesting complex gene 6	Photosynthesis-antenna proteins
*LOC_Os06g44010*	*ATWRKY40*	3.722	3.455 ± 0.418	WRKY DNA-binding protein 40	WRKY transcription factor
*LOC_Os12g02470*	*ATWRKY53*	6.512	5.360 ± 0.1837	WRKY family transcription factor	WRKY transcription factor
*LOC_Os11g02540*	*ATWRKY70*	4.669	2.862 ± 0.2069	WRKY DNA-binding protein 70	WRKY transcription factor
*LOC_Os07g04560*	*anac042*	1.566	2.641 ± 0.0344	NAC domain containing protein 42	NAC transcription factor
*LOC_Os10g42130*	*anac071*	2.630	2.798 ± 0.0182	NAC domain containing protein 71	NAC transcription factor
*LOC_Os09g36200*	*ATNYE*	1.437	1.375 ± 0.0246	Non-yellowing 1	NAC transcription factor
*LOC_Os01g21250*	*AtLEA5*	4.182	3.947 ± 0.0286	Senescence-linked gene 21	Biological process
*LOC_Os03g0529*	*SAG12*	1.043	3.139 ± 0.0128	Senescence-associated gene 12	Proteolysis (Biological process)
*LOC_Os06g16370*	*ATCOL2*	−2.334	−0.194 ± 0.0378	CONSTANS-like 2	GATA transcription factor
*LOC_Os03g08330*	*JAZ10*	1.336	0.893 ± 0.0239	Jasmonate-zim-domain protein 1	Plant hormone signal transduction
*LOC_Os01g09450*	*IAA26*	−3.341	−1.985 ± 0.0462	Phytochrome-associated protein 1	Plant hormone signal transduction
*LOC_Os02g57250*	*IAA13*	−1.316	−0.639 ± 0.0302	Auxin-induced protein 13	Plant hormone signal transduction
*LOC_Os01g28450*	*ATPR1*	6.095	5.015 ± 0.0108	Pathogenesis-related gene 1	Plant hormone signal transduction
*LOC_Os07g03730*	*PR1a*	10.420	4.106 ± 0.105	Pathogenesis-related gene 1	Plant hormone signal transduction
*LOC_Os01g09800*	*ATNPR1*	1.140	1.893 ± 0.1017	Pathogenesis-related gene 1	Plant hormone signal transduction
*LOC_Os04g32480*	*JAZ5*	3.746	2.084 ± 0.143	Jasmonate-zim-domain protein 10	Plant hormone signal transduction
*LOC_Os09g29660*	*ABCG11*	1.868343	1.901 ± 0.041	ABC super family	ABC transporter
*LOC_Os07g48260*	*ATWRKY54*	3.980698	4.035 ± 0.051	WRKY DNA-binding protein 54	WRKY transcription factor
*LOC_Os01g60700*	*ATSIK*	1.036138	1.241 ± 0.312	Serine/threonine protein kinase	Protein kinase superfamily protein
*LOC_Os04g33630*	*ATFD3*	−1.74014	−0.934 ± 0.063	Ferredoxin 3	Photosynthesis related
*LOC_Os09g36250*	*AtMYB42*	1.8693	1.012 ± 0.082	MYB domain protein 42	MYB domain transcripton factor
*LOC_Os04g43070*	*AMT1;1*	1.9946169	1.512 ± 0.101	Ammonium transporter 1;1	Ammonium transporter

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
