# Peer review of "Transcriptional Profile Corroborates that bml Mutant Plays likely Role in Premature Leaf Senescence of Rice (Oryza sativa L.)"

_ijms, 2019, doi:10.3390/ijms20071708_

Round 1
Reviewer 1 Report
The authors have included my comments into the revised manuscript, and provided sufficient explanation into the cover letter. Therefore, I have no objection whatsoever toward the acceptance of this manuscript.
Author Response
To Date: 02 April 2019
The Reviewer
International Journal of Molecular Sciences
Dear Reviewer,
First of all I would like to extend my sincere thanks to you for giving valuable time to review the manuscript and finalize it with substantial improvement. We have reviewed the manuscript according to your suggestion and try to address them in the manuscript. Please find the point-by-point responses as follows:
Response to Reviewer-1
Comments: The authors have included my comments into the revised manuscript, and provided sufficient explanation into the cover letter. Therefore, I have no objection whatsoever toward the acceptance of this manuscript.
Response: Thank you for giving your kind consent regarding acceptance our current manuscript for publication.
Therefore, I hope that you would be kind enough to take necessary action regarding the publication of the revised paper in this journal.
With Kind regards
Chunhai Shi (corresponding author)
Department of Agronomy
Address: Zhejiang University, Hangzhou 310058, China
Email: [email protected]
Phone: +86 57188982691
Fax: +86 57188982691

Reviewer 2 Report
The paper reports transcriptomic data from a bml mutant and WT. The manuscript is well presented. Some minor changes may be considered.
What was the rationale behind the pooling of 10 leaves per plant. By sure the 10 leaves were at different develomental stages. This may have masked the results
The conclusins are somewhat weak. Especially the last statement ·Therefore, we could
manipulate those genes either knock out mutation or CRISPR/Cas9 gene editing for ensuring delayed senescence for increasing duration of photosynthesis and higher yield in rice" is unrealistic in the view of the huge amount of differentially expressed genes identifies i the study
Author Response
To Date: 02 April 2019
The Reviewer
International Journal of Molecular Sciences
Dear Reviewer,
First of all I would like to extend my sincere thanks to you for giving valuable time to review the manuscript and finalize it with substantial improvement. We have reviewed the manuscript according to your suggestion and try to address them in the manuscript. Please find the point-by-point responses as follows:
Response to Reviewer-2
Comment: The paper reports transcriptomic data from a bml mutant and WT. The manuscript is well presented. Some minor changes may be considered.
Response: Thank you for your compliments and we try to make necessary changes for your consideration.
Comment: What was the rationale behind the pooling of 10 leaves per plant? By sure the 10 leaves were at different developmental stages. This may have masked the results
Response: We are sorry for writing mistake. Actually we have taken 10 leaves from 10 plants at same growing age as biological replicates. We have made necessary correction in the text of materials methods. Please see the lines 397 – 398.
Comment: The conclusions are somewhat weak. Especially the last statement ·Therefore, we could manipulate those genes either knock out mutation or CRISPR/Cas9 gene editing for ensuring delayed senescence for increasing duration of photosynthesis and higher yield in rice" is unrealistic in the view of the huge amount of differentially expressed genes identifies i the study
Response: Thanks for your observation. We try to make the conclusion more precise for strengthening the conclusion. Please see the lines 383 – 384.
Therefore, I hope that you would be kind enough to take necessary action regarding the publication of the revised paper in this journal.
With Kind regards
Chunhai Shi (corresponding author)
Department of Agronomy
Address: Zhejiang University, Hangzhou 310058, China
Email: [email protected]
Phone: +86 57188982691
Fax: +86 57188982691

This manuscript is a resubmission of an earlier submission. The following is a list of the peer review reports and author responses from that submission.
Round 1
Reviewer 1 Report
The authors provide annotation method with transcriptomics tools in order to observe the role of bml mutant in premature leaf senescence of rice. The bioinformatics pipeline involves several standard transcriptomics tools such as bowtie2, primer 5, and DESeq2. The provided annotations deserve attentions in this respect. However, there are several questions that should be answered in order to improve this manuscript, and the authors should revise it with incorporating the answer of these questions:
Line 183: Why most of the genes covered in biological process annotations?
Line 186: Did you find evidences of RNA binding as well?
Line 189: Figure 4: The graphic is not clear and too small. You should change it
Line 211: Is there any biological explanation why those KEGGS have the most significant enrichment?
Line 413: Did you use default parameters for Bowtie2. Please expose all the parameters that you used!
Line 415: What is the version of your reference genome? For example, human genome has version number as wel (eg.hg19)
Line 429: Why did you take 19 genes not more?
Line 430: Did you use default parameters for Primer 5? Please expose all the parameters that you used!
Reviewer 2 Report
This paper aims to provide new information on the genetic basis of senescence development for future applications in plant breeding.
Unfortunately, the experimental approach is not really adequate for this purpose. The brown mid-rib mutant obtained by the use of chemical mutagenesis is compared to wild type. RNAseq is used to define differenences in gene expression when the mutant plants just have initiated senescence symptoms.
There are two major critical points in this approach
The genetic differences between wild type and mutant are not estabkished. The mutagenesis treatment may have affected a large amount of genes not only those directly involved in the senescence process.
Sampling both mutant and wild type leaves at a single time point when the mutant plants already show initial symptoms of senescence is really not a sound approach. Induction of senescence reated gene start much before sensecence symptoms get visible. Your comparison reveales huge amount of differences in gene expression. This is to be expected . It is well established that during senescence many genes responsable for degradation processes are activated, including oxidative stress reated ones, while those related to active growth, including those reated to stimulating phytohormones are decreased.
This approach, however, does not give new information on the mechanisms of either premature or delayed sensescence .
Besides this fundamental criticism, in Material and Method section the authors need to specify in more detail the plant growth conditions and the age of the samples leaves.
The discussion is weak. The first part gives a decription of the phenotypic differences of the mutant in comparison to wild type which has already previously been published, followed by methodological explanation which should be shifted to Materials and Method section.
The rest discusses the data of the current work but, as evident due to the deficient experimental approach the discussion does not provide really new insghts into senescence regulatory processes. The conclusions are vague.